# Observation of dissociative quasi-free electron attachment to nucleoside via excited anion radical in solution

Jun Ma[1], Anil Kumar [2], Yusa Muroya[3], Shinichi Yamashita[4], Tsuneaki Sakurai[1], Sergey A. Denisov [5], Michael D. Sevilla [2], Amitava Adhikary[2], Shu Seki [1] & Mehran Mostafavi[5]

Damage to DNA via dissociative electron attachment has been well-studied in both the gas and condensed phases; however, understanding this process in bulk solution at a fundamental level is still a challenge. Here, we use a picosecond pulse of a high energy electron beam to generate electrons in liquid diethylene glycol and observe the electron attachment dynamics to ribothymidine at different stages of electron relaxation. Our transient spectroscopic results reveal that the quasi-free electron with energy near the conduction band effectively attaches to ribothymidine leading to a new absorbing species that is characterized in the UV-visible region. This species exhibits a nearly concentration-independent decay with a time constant of ~350 ps. From time-resolved studies under different conditions, combined with data analysis and theoretical calculations, we assign this intermediate to an excited anion radical that undergoes N1-C1′ glycosidic bond dissociation rather than relaxation to its ground state.

[1] Department of Molecular Engineering, Graduate School of Engineering, Kyoto University, Nishikyo-ku, Kyoto 615-8510, Japan. [2] Department of Chemistry, Oakland University, 146 Library Drive, Rochester, MI 48309, USA. [3] Department of Beam Materials Science, Institute of Scientific and Industrial Research, Osaka University, 8-1 Mihogaoka, Ibaraki, Osaka 567-0047, Japan. [4] Nuclear Professional School, School of Engineering, The University of Tokyo, 2-22 Shirakata Shirane, Tokai-mura, Naka-gun, Ibaraki 319-1188, Japan. [5] Laboratoire de Chimie Physique, UMR 8000 CNRS/Université Paris-Sud, Bât. 349, 91405 Orsay, Cedex, France. Correspondence and requests for materials should be addressed to J.M. (email: ma.jun.26m@st.kyoto-u.ac.jp) or to S.S. (email: seki@moleng.kyoto-u.ac.jp) or to M.M. (email: mehran.mostafavi@u-psud.fr)

Radiation-induced cellular DNA damage stems not only from the impact (i.e. direct effect) of primary high-energy photons and charged particles, but also from secondary species (excited molecules, free radicals, and free electrons) that are produced via radiolysis of cell components along the radiation tracks[1,2]. Secondary electrons are ubiquitous in an irradiated medium with an estimated quantity of $\sim 4 \times 10^4$ electrons per 1 MeV energy deposited[3]. They cause cascades of additional ionizations and excitations through inelastic scattering with molecules. As a result, low-energy electrons (LEEs) are generated with an excess kinetic energy of 0–20 eV[4].

DNA strand breaks, especially double strand breaks (DSBs), are the most important DNA damage that has been shown to lead to cell death and neoplastic transformation[1,5]. It is known that fully solvated electrons ($e_{sol}^-$) are ineffective at triggering DNA bond cleavage because they generally reside on biomolecules as stable anions[6]. For this reason, the conventional notion of electron-induced damage to the genome is mainly due to those electrons with sufficient energy to ionize or excite DNA, thereby leading to the formation of electron-loss radicals (holes) and excited states that cause subsequent molecular fragmentation[7]. In 2000 and onwards, the experimental observations from Sanche and coworkers showed that LEEs were able to cause single strand breaks (SSBs), as well as DSBs via dissociative electron attachment (DEA)[8,9]. This observation motivated a great number of mechanistic studies on the interaction of LEEs with DNA and its components in both the gas and condensed phases[10–15]. The low-energy resonance features in the yield of DSBs, SSBs, and anions produced by the impact of LEEs on model pyrimidine bases suggested that the initial step involves electron capture into the unoccupied molecular orbitals that are above the lowest unoccupied molecular orbitals (LUMOs) of the parent nucleobase, creating excited transient negative ions (TNIs*). Once the TNIs* are formed, they are shown to decay very rapidly leading either to a SSB via phosphate-sugar C–O σ bond cleavage[12,13] or result in unaltered base release via N1–C1′ glycosidic bond breakage[14,15]. Studies of DEA using various DNA models (monomers, oligomers with defined sequences, plasmid DNA) were often carried out under vacuum conditions; these experiments were limited to gas phase and condensed phase or micro-hydrated molecular targets[10–15].

In a polar medium (e.g. water), as shown in Fig. 1, LEEs successively lose energy to become quasi-free electrons ($e_{qf}^-$) and they can undergo multistep solvation prior to their complete localization as $e_{sol}^-$[2,16,17]. The transition from $e_{qf}^-$ to $e_{sol}^-$ is accompanied by the appearance of a strong optical absorption as the electron acquires a stable quantum state of binding energy, which was evidenced by time-resolved techniques, typically using a short pulse of high-energy electrons or a laser beam[16–18]. From the viewpoint of the action of LEEs, it is appropriate to suggest that a thorough understanding of the role played by short-lived non-equilibrated electrons would lead to a clearer picture of the basic mechanisms underlying the biological consequences of radiation. Therefore, a detailed knowledge of electron attachment to DNA/RNA in solution leading to the formation of the TNI* and the subsequent pathways of reactions that the TNI* undergoes, are of fundamental importance. However, these studies, even at a monomeric DNA-subunit (e.g., nucleosides, nucleotides) level, have been lacking. This may be due to challenges encountered in femtosecond laser spectroscopic investigations on the formation of TNI* and its reaction channels[19]. In contrast, the accelerator technique delivers a high-energy electron pulse to the solvent, and hence generates LEEs in accord with those in radiation biology and allows us to investigate the chemistry induced by radiation-produced electrons in liquids[19].

Unfortunately, under ambient conditions, as the solvation of LEEs in water is fast ($\leq 1$ ps), and the limitation of current high-

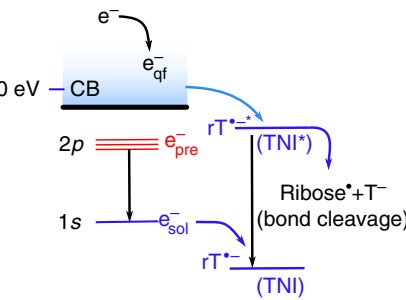

**Fig. 1** A schematic diagram of energy level showing different states of electrons during trapping and relaxation. These processes take place in a polar medium following ionizing radiation in the presence of ribothymidine (rT). An excess electron in the conduction band (CB), representing a quasi-free electron ($e_{qf}^-$), eventually becomes trapped ($e_{sol}^-$) in the solvent cage. The excited state of $e_{sol}^-$ is considered as a "presolvated" electron, $e_{pre}^-$. Electrons captured by solute molecules produce transient negative ions (TNI or rT•−). The TNI in its excited state (TNI*) can either liberate the excess energy to the solvent (relaxation) or undergo bond breaking (dissociation)

energy electron pulse (7 ps) radiolysis system[19] prevents investigation of the complete time resolution of the $e_{pre}^-$ solvation versus its attachment processes. Relaxation of the electron (i.e. from $e_{qf}^-$ to $e_{pre}^-$ and ultimately to $e_{sol}^-$, Fig. 1) can be viewed as a multistep transition from the delocalized conduction band with p-like excited states to s-like ground states[2,16,20]; however, experimental characterization of the specific state of electron that is required for the DEA processes still remains elusive. In addition, formation of the excited states of DNA anion radicals via electron attachment has been suggested but has never been observed[1,19].

It is known that the time of solvation of electrons in alcohols is of the order of several ps[18,20]. Here, we observe that electron solvation events in diethylene glycol (DEG) are relatively slow and occur on the order of tens of picoseconds. As the value of the dielectric constant ($\varepsilon_r = 31.69$) of DEG is much closer to that of a biological cell[21], it allows us to investigate the reactions of electrons that are more relevant to biological system. Also, DNA can retain its native double-stranded structure and biological activity in glycol solutions[22]. We solely find that ribothymidine (rT), a DNA/RNA model, can be sufficiently dissolved in DEG (up to 0.5 M) to scavenge electrons on a short time scale. We conduct picosecond pulse radiolysis studies of rT solutions in DEG at different concentrations (0–0.5 M) under ambient conditions, that allow us to observe real-time dynamics of the electron attachment to DNA-components at certain stages of electron relaxation. Our pulse radiolysis technique[23] is based on a picosecond laser-triggered electron accelerator coupled with transient UV–Vis/IR absorption spectroscopy (Supplementary Figure 1). The key transient species ($e_{pre}^-$, $e_{sol}^-$, and TNI* of rT•−*) can be directly observed by the probe light covering a broad wavelength range from 370 to 1100 nm. In addition, theoretical calculations using density functional theory (DFT) show that rT•−* undergoes N1–C1′ glycosidic bond dissociation rather than relaxation to its ground state.

## Results

**Electron solvation in liquid DEG**. The transient absorption spectra of neat DEG is shown in Fig. 2a. A significant fraction of electrons formed in DEG immediately after the electron pulse have undergone relaxation. A transient feature absorbing above 900 nm rapidly diminishes, and a broad intense signal with a maximum at around 750 nm undergoes a continuous blue shift of

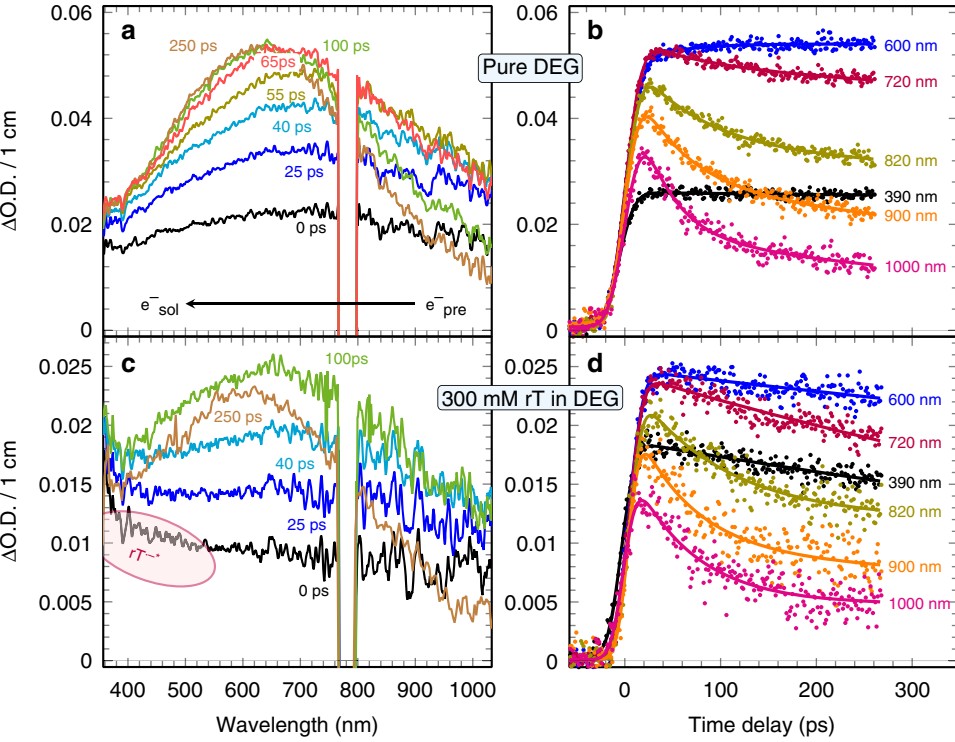

**Fig. 2** Transient absorption profiles of pure diethylene glycol and ribothymidine solutions under ambient conditions. Spectra (**a**, **c**) and kinetics (**b**, **d**) for neat DEG and 0.3 M ribothymidine (rT) solutions, respectively. The region at 780 nm is filtered out. The duration of the electron pulse is ~7 ps. The dose per electron pulse is 25.3 Gy

ca.120 nm, accompanied with a growing amplitude of about hundreds of picoseconds (e.g. 250 ps, Fig. 2a). At the end of the spectral evolution (Fig. 2a), a broad and featureless absorption band builds up with a peak at 630 nm. Based on previous studies in various alcohols[24], this band is assigned to the spectrum of $e_{sol}^-$. Quenching of the signals by adding acetone (an electron scavenger) confirms this assignment and nanosecond pulse radiolysis measurements indicated that the lifetime of $e_{sol}^-$ in neat DEG is around 5 µs. Alcohol radicals are formed by ionization, but they absorb only in the UV region below 300 nm. As a result, the time-dependent spectral changes in Fig. 2a clearly show that the electron solvation essentially consists of at least two distinct states, one absorbing in the near-infrared, and one in the visible range.

Figure 2b presents kinetic traces of the transient electrons in DEG at various wavelengths (390–1000 nm) with a non-monotonic kinetics and with no isosbestic point. Figure 2b clearly shows that the signals at 390 and 600 nm rise fast, then remain almost constant for the next hundreds of ps, and the one at 900–1000 nm follows a continuous decrease. The overall behavior of transient spectra and kinetics show a stepwise transition between the two above-mentioned states ($\tau_1$) along with simultaneous blue shift of visible state ($\tau_2$). To gain further insights into the dynamics, the best multi-exponential global fits to transients at a variety of wavelengths find two components, a fast component with a time constant of $\tau_1 = (45 \pm 15)$ ps and a slow component with $\tau_2 = (80 \pm 30)$ ps. Our results are in agreement with the previously reported spectral behaviors of electrons in liquid water[25,26], methanol[27,28], ethylene glycol[29], and polyols[30] observed by femtosecond laser spectroscopy.

In alcohols, the formation time of $e_{sol}^-$ was reported to be much slower than in water and it is determined by many factors, such as the size of alcohol, the number of the hydroxyl group and dielectric relaxation time for molecular rotation. Our model that uses multi-exponential global fits to account for transients involved in electron solvation processes, agrees well with the hybrid model proposed by Pépin et al.[25,27] and Barbara et al.[26,28]. According to this model and theoretical simulations[31–33], the faster relaxation component, $\tau_1$, can be assigned to the transition of the weakly bound *p*-like states, and the slower continuously shifting component, $\tau_2$, is attributed to $p \rightarrow s$ radiationless decay or cooling of a vibrationally excited *s*-state. Recent liquid-jet photoelectron spectroscopy results supported the latter view[34,35]. For instance, angular-resolved photoemission measurements showed the photo-electrons generated from the state of the slow component is associated with an isotropic character, revealing the existence of the "hot" ground state[35]. Therefore, our results on the precursors of $e_{sol}^-$ and their equilibration process in DEG represent a starting point for further investigation of the electron transfer process to DNA constituents.

**Electron attachment to rT leading to TNI\* formation**. To investigate the electron attachment to rT and the consequent formation of rT•–\*, picosecond pulse radiolysis studies were performed in rT solutions (50–500 mM) in DEG. In these solutions, direct ionization or excitation of solute itself is not significant. As an example, the transient spectra and the kinetics at different wavelengths are reported for the concentration of 300 mM in Fig. 2c, d, and the effect of the rT concentration on the kinetics are shown in Fig. 3. Rapid electron capture by rT generates a new transient signal that is immediately and clearly visualized in the UV–visible regions shown in Fig. 2c. In contrast to the slight increase in neat DEG, the transient kinetics in the UV–visible region (370–600 nm) shown in Fig. 2d and Fig. 3a, b of rT solutions display an obvious decay. From Fig. 3c, d, we observe a substantial decrease of the initial near-infrared absor-bance that correlates exponentially with increasing rT

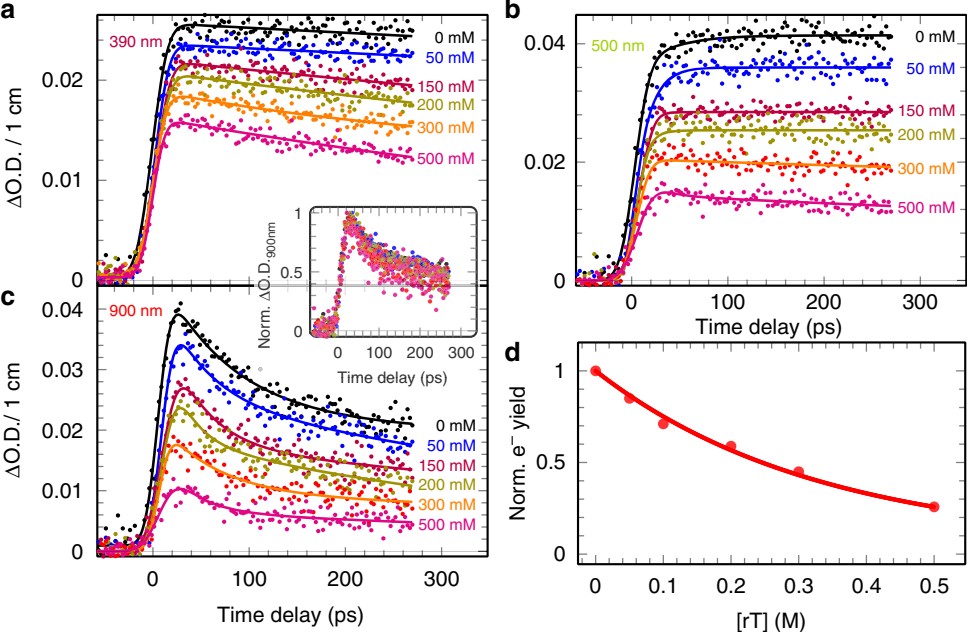

**Fig. 3** Transient absorption kinetics at various ribothymidine concentrations. Kinetics at wavelengths: **a** 390 nm, **b** 500 nm, and **c** 900 nm. The inset shows the normalized kinetics at 900 nm at different concentration of ribothymidine (rT). Electron solvation is shown to be complete within about 300 ps and the geminate recombination between the electron and hole can be neglected at this timescale. **d** Relative electron yield in the presence of rT

concentrations. These results clearly show that rT has scavenged a significant fraction of the electrons prior to their being trapped in DEG. For instance, about 12% and as high as 75% of electrons are captured by rT molecules at the concentration of 0.05 and 0.5 M, respectively. Analyses of normalized kinetics at the infrared region from 720 to 1000 nm, at which the absorptions are only associated with $e_{pre}^-$, are found to be nearly identical (Fig. 3d, inset) with those observed in neat DEG (Fig. 2b). Also, the global fitting for the transient in each rT solution at the higher wavelength (>720 nm) shows that the characteristic times of two components ($\tau_1$ and $\tau_2$) remain almost unchanged. Therefore, these results establish that even such a significant extent of electron scavenging takes place before electron localization in pre-existing traps, the presence of rT does not affect significantly the subsequent electron solvation process in DEG (Fig. 3); also, rT does not react with $e_{pre}^-$ on the time scale of hundreds of ps. Based upon our assignments of the states of $e_{pre}^-$ and $e_{sol}^-$, we conclude that the electrons captured by rT within the pulse duration (≤7 ps) are most likely attributed to $e_{qf}^-$ at or above the conduction band. Because energy levels of trapped electrons in a solvent strongly correlate with time of electron solvation[36–38], it is emphasized that a small but non-negligible energy barrier prevents the ultrafast attachment by electrons lying at lower energy trapping sites, such as p-like states (component $\tau_1$), vibrationally excited s-like state (component $\tau_1$), and ground s-like state ($e_{sol}^-$). Indeed, the complete relaxation of electrons starts from delocalization of excess electrons possessing a high mobility (>1 cm$^2$ V$^{-1}$ s$^{-1}$)[20]. In water and alcohols, the vertical binding energy (VDE) of a conduction band electron is much lower than that of its weakly bound states and ground states[37–40]. Moreover, first-principle molecular dynamics simulations predicted that excess electrons attach rapidly (<15 fs) to solvated DNA nucleotides[41], which supports our experimental results. Therefore, based on these studies and on our results (Figs. 2 and 3), the higher chemical reactivity of $e_{qf}^-$ relative to that of trapped electrons is justified. In addition, our findings establish the quantum states of electrons that are involved in formation of TNI* via ultrafast electron attachment to DNA-components in solution.

As described above, the presence of rT has only changed the initial absorbance of $e_{pre}^-$ and the observed IR spectra of $e_{pre}^-$ are identical with that in DEG. Consequently, the transient profiles of rT intermediates were obtained as shown in Fig. 4, via a combination of a multivariate curve resolution alternating least squares (MCR-ALS) analysis of a data matrix (Supplementary Figure 2) previously described[42] and via simply subtracting the $e_{pre}^-$ absorption in neat DEG (Supplementary Figure 3). In studied wavelength range, the spectra of this species are characterized as a mono-peak absorption band extending to the UV (Fig. 4a) and it does not evolve with the delay time (Supplementary Figure 3). Figure 4a inset also showed that this species undergoes a decay that is nearly independent of the concentration. The lifetime of this species is estimated to be ~350 ps based on the lineally fitting of the kinetic in logarithm, as well as the half-life of TNI* shown in Fig. 4b, which agrees with previously measured rates of charge-induced dissociation[43].

In the liquid phase, electron attachment to rT results in the formation of excited rT•−* (i.e., TNI*) or ground state, rT•−, i.e., TNI. rT•−* will subsequently, either dissociate to a neutral radical (R•) and an anion when the energy is accessible or relax to a stable anion radical by liberating energy to the solvent. This latter species, rT•−, is, essentially, identical in nature with that from the reaction of the fully $e_{sol}^-$ (see Fig. 1). It is less likely that the transient kinetics correspond to the decay of fragment radicals R•. This is because in DEG (viscosity = 35.7 cP; 25 °C), R• should react with other radicals through nearly diffusion-limited reactions on timescale of tens of nanoseconds. In addition, the decay rate should be affected by the concentration of R•. More importantly, the initial absorbance of the new species at various concentrations is linearly correlated with the number of electrons captured (Supplementary Figure 4). After ruling out this possibility, the key question now is whether it corresponds to the excited anion radicals or fully solvated anion radicals or both. To answer this question, the MCR-ALS analysis of a data matrix showed several important features of species involved as displayed in Fig. 4: (i) only two absorbing species ($e^-$ and rT•−*) exist at the early time. rT•−* is formed within the pulse and not

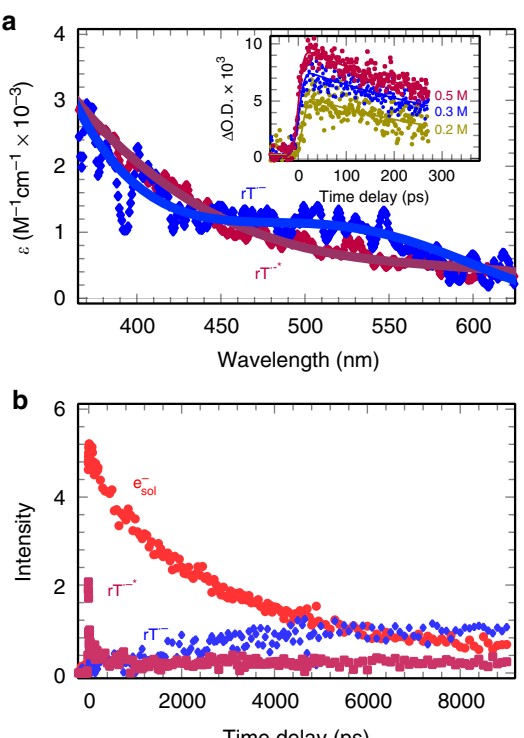

**Fig. 4** Deconvolution of absorbing species in a data matrix. **a** Spectra of excited and ground states of ribothymidine anion radicals, rT•−* and rT•−, respectively. The inset shows the kinetics of rT•−* at different concentrations. **b** Deconvolution of transient data using a multivariate curve resolution alternating least squares (MCR-ALS) method representing the kinetics of rT•−*, rT•−, and the solvated electron (e$_{sol}$-)

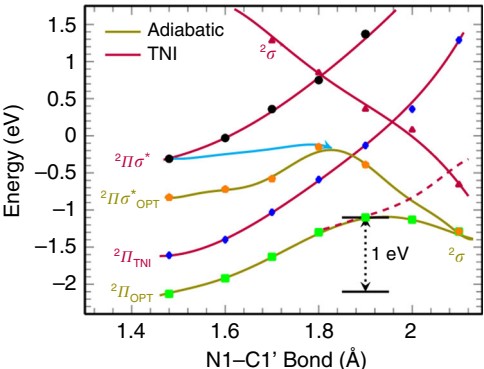

**Fig. 5** Potential energy surface profiles for N1–C1′ bond cleavage. Shown for the ground and vertical excited states of rT anions (TNI*). The energies were scaled by adding −1.6 eV to the actual calculated values (Supplementary Figure 8) to match with the theoretical adiabatic electron affinity (2.1 eV) of rT. The light blue solid line is the proposed path leading to barrierless fast N1-C1′ bond dissociation

after the electron pulse (ii). rT•−* decays in a few 100 ps and the decay of the electron during this time is almost negligible (Fig. 4b). (iii) The decay of the rT•−* does not form the rT•− because the formation of the latter only correlates with the decay of e$_{sol}$- at longer time (Fig. 4b). Besides, to compare the spectral difference (the band shape, lifetime, and extinction coefficient) between rT•−* and rT•−, we performed additional measurements of DEG solution at low concentrations (1–10 mM), in which e$_{qf}$- and e$_{pre}$- are not scavenged by rT at long range and all of them are converted into rT•− (Fig. 1). The resulting anion radicals rT•− at microsecond timescale show a distinct transient spectrum (blue curve in Fig. 4a) and remain stable over a long period of time (lifetime >10 µs, see Supplementary Figure 5). These results conclude that the transient signals are attributed to the excited anion radical rT•−* formed from e$_{qf}$- attachment to rT in liquid DEG.

In contrast to the results obtained in DEG solutions, studies using aqueous rT did not provide any evidence of the bond breaking. All TNIs* relaxed to rT•− (Fig. 1), which is in the agreement with our previous work[19]. For a given rT concentration, e.g., 0.5 M, the scavenging time of e$_{pre}$- is about 400 fs both in water and in DEG as the rate constant of e$_{pre}$- scavenging by rT is found to be similar, ~5 × 10$^{12}$ M$^{-1}$ s$^{-1}$. However, as the solvation dynamics in water is much faster, the time for excess electrons reaching into the pre-existing traps of liquid water is on the order of tens of fs and the lifetime of p-like states of electron was around 300 fs. Thus, rT in liquid water cannot react with the higher electronic states of electrons, such as p-like states or conduction states, which, in turn, do not offer enough energy for the bond rupture of TNIs*. Thus, our results clearly point out the energy states of a single electron and pre-existing traps in the

solvent medium are the decisive factor for the occurrence of TNIs* fragmentation.

LEEs-induced decomposition of thymidine was first studied by analysis of irradiated products at the molecular level in the condensed phase[14]. These experiments showed the effective excision of the thymine base from low-energy electron-irradiated DNA oligomers of defined sequences through N1–C1′ glycosidic bond cleavage. In the mechanistic study, Ptasinska et al.[15] suggested that the resonant localization of an electron with an energy as low as 1.2 eV on the sugar moiety in the gas-phase can cause bond cleavage. Kočišek and co-workers[44] recently provided better insight into the role of water played in DEA using a "bottom-up" approach in a molecular beam, by progressively micro-hydrating the target molecule. This study highlights the suppression of dissociation of N1–C1′ glycosidic bond by increasing the hydration degree of thymine and uracil. Thus, the decay of rT•−* observed in our time-resolved experiments that we have assigned to a dissociative process could be associated with the dynamics of glycosidic bond cleavage[14].

**Modeling of excited TNI surfaces leading to N1–C1′ bond breakage.** It is well-known that LEEs on interacting with a molecule create TNI resonances which are equivalent to vertical excited states of the electron adduct of the parent molecule[45–49]. Based on this understanding, calculations of the transition energies to vertical excited states of a TNI can predict the specific resonance energies available for direct capture of LEEs. In this work, the transition energies of TNI of rT in DEG (ε$_r$ = 31.69) are calculated using the time-dependent DFT (TD-DFT) implemented in Gaussian 16[50]. The complete methodology is abbreviated as TD-ωB97XD-PCM/6-31G**. Use of a compact basis set (6–31G**) in these calculations avoids mixing of resonances with the continuum[46–48,51].

From the nature of the TNI potential energy surface (PES) ($^2π_{TNI}$, Fig. 5), we see that as the N1–C1′ bond elongates the energy of ground state of the TNI ($^2π_{TNI}$) surface increases until it crosses the dissociative $^2σ*$ surface with a barrier of 1.6 eV. The energy of the first excited state $^2π → ^2πσ*$ ($^2πσ*$ surface) also increases as N1–C1′ bond elongates and at 1.8 Å it crosses the dissociative $^2σ*$ surface having a barrier of ca. 1 eV. We also optimized the $^2πσ*$ excited state (The $^2πσ*$OPT was not fully optimized because during optimization this excited state switches to another excited state after few cycles of the optimization steps. Thus, we use the energy of $2πσ*$OPT state just before the switch

to the other excited state.) designated as $^2\pi\sigma^*_{OPT}$ in Fig. 5. As expected, the adiabatic state, $^2\pi\sigma^*_{OPT}$, lies lower than the vertical $^2\pi\sigma^*$ surface and after surpassing a small barrier of ca. 0.6 eV, the N1–C1′ bond dissociates. The light blue solid line in Fig. 5, denotes a proposed barrier-free dissociation path which occurs on capture of a quasi-free electron into the vertical πσ-MO of rT ($^2\pi\sigma^*$) which upon extension of the N1–C1′ bond, relaxes from the vertical to the adiabatic surface.

The ground state adiabatic PES of rT anion radical ($^2\pi_{OPT}$, Fig. 5) shows that the N1–C1′ bond dissociation involves a substantial barrier ca. 1 eV as reported earlier[45]. From the overall nature of the PESs, we inferred that the energy of the lowest excited state of the TNI is located at −0.3 eV ($^2\pi\sigma^*$) and provides the most likely path for immediate dissociation of rT via electron attachment. To mimic the experiment, we scaled the energy in Fig. 5 by adding −1.6 eV to the calculated values for matching the adiabatic electron affinity (AEA) (2.1 eV) of rT that were calculated using the ωB97XD-PCM/6-31++G** method with actual energy values presented in the supporting information (Supplementary Figures 6–9). From Fig. 5, it is evident that the vertical $^2\pi\sigma^*$ excited state of TNI has energy of −0.3 eV which lies within the conduction band energy range. An overview of the energies of the electron and TNI of rT in DEG is shown in Fig. 1 and Supplementary Figure 9. Thus, electrons generated in the conduction band should be efficiently captured into the excited state πσ*-MO of the rT TNI and proceed via gradual relaxation of the structure on bond elongation to a barrier-free N1–C1′ glycosidic bond cleavage leading to thymine release.

## Discussion

Our spectroscopic observations in DEG establish that the quasi-free electrons form two localized electron-solvent configuration states (in the infrared region and in the visible region within the timescale of the electron pulse (<7 ps)); the former is characterized as a $p$-like state and the latter is assigned to a vibrationally hot ground state which gradually relaxes to form a solvated electron $e_{sol}^-$. In the presence of rT, our results show that dissociative electron transfer occurs only by the quasi-free electrons at or above the conduction band rather than via $e_{pre}^-$ and $e_{sol}^-$. The resulting excited rT anion radical rT•−* observed on a picosecond timescale has been fully characterized, and it exhibits a spectrum that is different from the spectrum of the stable anion radical rT•− which is observed on the microsecond timescale in dilute DEG solutions at room temperature. The combination of time-resolved results and DFT calculations establishes that the observed transients rT•−* can be attributed to the excited state πσ*-MO of the TNI of rT, and its dissociation proceeds via gradual relaxation of the structure on bond elongation through a barrier-free N1–C1′ glycosidic bond cleavage. Our results further imply the generation of biomolecular damage does not necessarily require electrons carrying kinetic energy. In cellular systems the water molecules have inherently long relaxation times. Conduction band electrons ($e_{qf}^-$) formed in cells should have longer lifetimes than those found in water or in DEG; as a result, these longer-lived $e_{qf}^-$ would contribute to biomolecular damage. Thus, the insights gained in our present study could pave the way to directly investigate the long-standing mystery of electron-driven reactions in radiation chemistry and biology.

## Methods

**Pulse radiolysis experiment**. The chemical compounds (rT and DEG; purity, >99%) were purchased from Sigma-Aldrich and used without further purification. The 7 ps pulse radiolysis coupled with transient absorption measurements were performed at the electron facility LINAC (Tokyo University) coupled to a transient absorption broadband probe spectroscopy (380–1050 nm) instrument. Additional time-resolved radiolysis results were also carried out at ELYSE facility (Paris-Saclay University) for comparison. The experimental data matrices were analyzed by a

MCR-ALS approach as previously described[52]. Details about experimental apparatus, methodologies, and data analysis are provided in Supplementary Notes 1 and 2.

**TD-DFT calculations**. In this study, the ωB97XD density functional and 6–31G** basis set were used and the effect of bulk solvent with dielectric constant of DEG ($\varepsilon = 31.69$) was incorporated via the use of the integral equation formalism polarized continuum model (IEF-PCM). Details about the TD-DFT calculations are provided in Supplementary Note 3.

## Data availability

The data that support the findings of this study are available from the corresponding authors upon request.

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

## Acknowledgements

This work was partially supported by Grant-in-Aid for Scientific Research Nos. 17F17085, 15K21721, 15H04243, and 18H03918) from Japan Society for the Promotion of Science (JSPS). The authors thank Dr. Mitsumasa Taguchi (Takasaki Advanced Radiation Research Institute, National Institute for Quantum and Radiological Science and Technology, QST) for γ-ray irradiation as well as the Shared-Use Program of QST. J.M. thanks the Japan Society for the Promotion of Science for a Fellowship. A.K., M.D.S., and A.A. thank the National Cancer Institute of the National Institutes of Health (Grant R01CA045424) for support. A.A. is also grateful to Research Excellence Fund (REF) from Center for Biomedical Research (CBR) at Oakland University for support. It is our pleasure to thank Prof. M. Uesaka and Mr. T. Ueda (The University of Tokyo) for providing support with the accelerator experiment. We thank Dr. Tamura (Kyoto University) for providing support with the HPLC measurements. The authors are grateful to Prof. David Becker (Chemistry Department, Oakland University) for copy-editing the revised manuscript.

## Author contributions

This work resulted by the collaboration among J.M., M.M., S.S., and A.A.; J.M., M.M., and S.S. designed the experimental part of the project; J.M., Y.M., and S.Y. performed key experiments; T.S., S.A.D., S.S., and M.M. performed additional necessary experiments; A.K. and M.D.S. performed TD-DFT calculations and analyzed data; J.M., S.A.D., and M.M. analyzed the experimental data; A.A., J.M., S.S., A.K., M.D.S., and M.M. wrote the manuscript.
