## [Peer Review File · Nature Communications]

Reviewers' comments:

Reviewer #1 (Remarks to the Author):

The present manuscript introduces the last work of Jun Ma et al., which is part of a set of articles (1,2) published recently on the reaction of radiation-induced electrons in different states of solvation reacting with basic DNA compounds, i.e. nucleosides, in liquid solution. I read with considerable interest the manuscript. The objective of this research is to demonstrate that the fundamental process of radiation damage known as dissociative electron attachment (DEA) exists in the liquid phase. DEA is one of the most important and basic reaction induced by the secondary electrons produced in large quantities by any type of ionizing radiation. It has been investigated in details for more than half a century with gaseous molecules and more recently within molecular solids and at surfaces. The process has many ongoing and potential applications, including radiotherapy and concomitant chemo-radiation therapy. Although the phenomenon lies at the basis of important advances in many fields, it has never been demonstrated to occur in the liquid phase. This information is clearly missing from our present knowledge and slowing down significant progress, particularly in cancer treatment via radiotherapy. For this reason, the authors have chosen to try to demonstrate that DEA occurs in a liquid, containing as solute biomolecules, which constitute a subunit of DNA, the target in radiotherapy. There is no doubt that their work is impressive and highly significant. On the other hand, due to the importance of such results, it is important that the authors clearly demonstrate in their manuscript the observation of DEA in diethylene glycol (DEG) before publication. There are major flaws that require to be seriously addressed before publication in Nature Communications. My report below provides a number of suggestions to do so, with some criticism on the presentation of their results.

The authors used a 15-picosecond pulse of high energy (18 MeV) electrons to time-resolve the generation of secondary low-energy electron (LEE) in liquid DEG and determine the electron attachment dynamics to ribothymidine at the stages of electron relaxation via absorption spectroscopy. Ideally, such experiments should be done in water, which is a more biological solvent, but the relaxation time of the electron is too fast for real-time observation of the solvation steps. However, in cells the lifetime of LEEs should be slower than in pure water and closer to that in DEG. Their spectroscopic results reveal that the quasi-free electron with energy either higher or at the level of conduction band, attaches to ribothymidine. The fast electron capture by ribothymidine leads to a new species characterized in the UV-visible region. Their time-resolved studies are complemented by DFT calculations and analyses of molecular products formed by gamma radiolysis. Considering all the information obtained, the authors assign the transient intermediate state as the excited ribothymidine anion radical that undergoes exclusively an N1-glycosidic bond dissociation rather than relaxation to its ground state.

1. Experimental results retrieved from the picosecond pulse radiolysis setup.

Within the framework of pump-probe time-resolved absorption spectroscopy, it is well known that measuring appropriately the absorbance of transient species within a spectral range located around and below 400 nm is more than very difficult using the white light generated from sapphire as the probe, mainly because of the very poor quality of probing light that it produces below 420 nm (3). Furthermore, this point is of particular importance here as the absorbance retrieval at 390 nm is critical to support the authors' line of reasoning. Can the authors explain how they addressed this critical issue to make sure that their results can be trusted?

Such implementation of their pump-probe setup is surprising as the authors seem to be aware of this problem, since they used CaF₂ for the white-light generation, in their previous study (1,2), which is known to appropriately cover the near-UV range.

2. Transient absorption spectra analysis.

According to the author's own writing on their previous work (2), the spectra of the type of system studied in the present work are known to strongly overlap; as a result, it becomes difficult to deconvolute individual spectrum. To alleviate this problem, they previously used a global data analysis approach, which seemed very rigorous (section 1 of the SI of (2)). In the present study, the authors did not specify if the same approach was applied. Is it correct to conclude that the transient spectra of interest in the present work were retrieved by using a more trivial data analysis method, which appears much less rigorous? If it is actually the case, can the authors explain why they failed in applying the same rigorous data analysis protocol as in (1, 2)?

3. Transient absorption kinetics of the TNI.

According to author's line of reasoning and the results displayed in Figure 3, the absorption kinetics at 525 nm appear to be of particular interest. Unfortunately, such kinetics are not displayed and properly investigated, considering that the kinetic of absorbance at 525 nm as a function of time would have provided better insight on the fate of TNIs*, i.e., exclusive decomposition via dissociation of the C-N glycosidic bond versus decay toward TNIs ground state, by analyzing the kinetics at 390 nm versus that at 525 nm. Furthermore, it might have been possible then to evaluate the fraction of excited TNIs that decays toward the ground state of TNIs versus the fraction of TNI* that results in the dissociative process.

Note that, as regards to the dependency of the kinetics displayed in figure 3.b on the concentration of nucleoside, the authors should furnish the characteristic time extracted from the exponential fit, together with standard deviations, and let the reader judge if it is independent of ribothymidine concentration.

In figure 1.d, why the kinetics at 1000 nm is not displayed? In figure SI 3, it seems that the absorbance at 1000 nm of 0.3 M ribothymidine / DEG increases relatively to neat DEG at early time (10 ps). Can the authors provide some explanation / interpretation about such results?

Throughout the manuscript and supporting information, display of absorption spectra and kinetics with various different absorbance Y scale / unit, is very confusing and should be made coherent. For example, the values of absorbance associated with the kinetic in 0.3 M riboT displayed in figure 3.b

appear inconsistent with the values of absorbance associated with the spectra displayed in figure SI 3.d. While the value of absorbance at 200 ps in figure 3.c matches the one displayed in figure SI 3.d, the absorbance at 40 ps in figure SI 3.d doubles relatively to figure 3.b. The authors need to deal with of such inconsistencies.

4. Identification of unaltered base (thymine) release via electron-mediated glycosidic bond cleavage in DEA to TNI* of ribothymidine.

First, the authors should definitively furnish the mass spectrum of irradiated ribothymidine solution at low concentration (20 mM) to let the reader judge of the absence of thymine.

Then, the authors should properly acknowledge the recent study published by Fedor and co-workers (4) that highlights the different outcome of the DEA on hydrated DNA compounds that, unlike DNA compounds under vacuum condition, decompose exclusively via dissociation of the C-N glycosidic bond.

Did the authors made sure that the production of thymine is linear as a function of dose between 0 and 300 kGy? This is of a particular importance to ensure that the base release reaction is a first order process.

The results presented here seems to be based on the assumption that the base release is specific to DEA. However, the release of unaltered bases is a general phenomenon of the free-radical-induced reactions of DNA and its constituents (section 10.8.7 of (5)). By the authors own admission, in the radiolysis of a solution, there occurs numerous reactions between radicals at different timescales from femtoseconds to seconds. The fact that they do not observe the production of thymine at low concentration (20 mM) seems to support their initial hypothesis. But, can the authors explain why the base release induced by the possible reaction of free radicals, e.g. on the sugar moiety, is dismissed?

In my opinion, the author's line of reasoning would be strengthened by providing the quantitative evaluation of thymine production as a function of the concentration of nucleoside to confirm that the kinetic of such production is compatible with the hypothesis of a specific involvement of the DEA process in the base release reaction, and thus establish a stronger correlation between the formation of TNI* and the thymine production.

Minor comments:

I suggest that the word "direct" be deleted from the title. After all the data manipulation to measure the dynamics of the states of the electrons, most readers may not see these measurements as very direct. On the other hand, the authors may want to mention in the title that the measurements were performed in the liquid state.

I generally found the manuscript to be difficult to read. The authors should carefully review the next version of the manuscript. The authors need also to be careful about typographical errors, e.g. errors in the legend of figure SI 3.c, in the Y-label of figure SI 3.d, in the conclusion of the manuscript.

Line 51 : write “in gas- and condense-phase”

Lines 52-55 The sentence “The low-energy resonance features in the yield of DSBs, SSBs, and anions produced by the impact of LEEs on model pyrimidine bases suggested that the initial step involves electron capture into the LUMOs (lowest unoccupied molecular orbitals) of nucleobases, creating excited states of transient negative ions (TNIs*)” is ambiguous. Electron capture in lowest unoccupied molecular orbital cannot produce a TNI in an excited state, but they write “orbitals”. How many orbitals are considered lowest?

Lines 61-64: Here the authors introduce the different states of the electron with the sentences “In a polar medium (e.g., water), the most relevant phase of biology and chemistry, LEEs successively lose energy to become quasi-free electrons (eqf^-) and they can undergo multistep solvation prior to their complete localization as $esol^-$. The transition ($epre^-$) from eqf^- to $esol^-$ is accompanied by the appearance of.....”. They refer to ($epre^-$), eqf^- and $esol^-$, but these symbols are not defined yet. I suggest that they first define the different states of the electron with their energy diagram of Scheme. 1, including that of LEEs.

Recommendation: Publication in Nature Communications, after the authors have adequately answered all comments in this report.

References

- (1) J. Ma, F. Wang, S. A. Denisov, A. Adhikary, and M. Mostafavi, *Science Advances* 3 (2017), 10.1126/sciadv.1701669, <http://advances.sciencemag.org/content/3/12/e1701669.full.pdf>.
- (2) J. Ma, S. A. Denisov, J.-L. Marignier, P. Pernot, A. Adhikary, S. Seki, and M. Mostafavi, *The Journal of Physical Chemistry Letters* 9, 5105 (2018), PMID: 30132673, <https://doi.org/10.1021/acs.jpcllett.8b02170>.
- (3) M. Bradler, P. Baum, and E. Riedle, *Applied Physics B* 97, 561 (2009).
- (4) J. Kocisek, B. Sedmidubská, S. Indrajith, M. Farnik, and J. Fedor, *The Journal of Physical Chemistry B* 122, 5212 (2018), PMID: 29706064, <https://doi.org/10.1021/acs.jpccb.8b03033>.
- (5) C. von Sonntag, *Free-Radical-Induced DNA Damage and Its Repair* (Springer-Verlag Berlin Heidelberg, 2006).

Reviewer #2 (Remarks to the Author):

The authors use pico-second laser-driven electron pulse radiolysis to study dissociative electron attachment (DEA) to selected DNA components in real liquid environments under ambient conditions. They determine that the intermediate is an electronically-excited anion which rapidly decays by DEA into products. On the one hand, I do not find this especially surprising, but on the other hand, this appears to be the first actual demonstration of that process in a real liquid. While parts of this manuscript are well written, other parts would be improved if a native English speaker were to edit it. On balance I like this manuscript and feel that it presents important new findings. I support its publication in Nature Communications.

Response to reviewers' comments

Reviewer: 1

The present manuscript introduces the last work of Jun Ma et al., which is part of a set of articles (1,2) published recently on the reaction of radiation-induced electrons in different states of solvation reacting with basic DNA compounds, i.e. nucleosides, in liquid solution. I read with considerable interest the manuscript. The objective of this research is to demonstrate that the fundamental process of radiation damage known as dissociative electron attachment (DEA) exists in the liquid phase. DEA is one of the most important and basic reaction induced by the secondary electrons produced in large quantities by any type of ionizing radiation. It has been investigated in details for more than half a century with gaseous molecules and more recently within molecular solids and at surfaces. The process has many ongoing and potential applications, including radiotherapy and concomitant chemo-radiation therapy. Although the phenomenon lies at the basis of important advances in many fields, it has never been demonstrated to occur in the liquid phase. This information is clearly missing from our present knowledge and slowing down significant progress, particularly in cancer treatment via radiotherapy. For this reason, the authors have chosen to try to demonstrate that DEA occurs in a liquid, containing as solute biomolecules, which constitute a subunit of DNA, the target in radiotherapy. There is no doubt that their work is impressive and highly significant. On the other hand, due to the importance of such results, it is important that the authors clearly demonstrate in their manuscript the observation of DEA in diethylene glycol (DEG) before publication. There are major flaws that require to be seriously addressed before publication in Nature Communications. My report below provides a number of suggestions to do so, with some criticism on the presentation of their results.

The authors used a 15-picosecond pulse of high energy (18 MeV) electrons to time-resolve the generation of secondary low-energy electron (LEE) in liquid DEG and determine the electron attachment dynamics to ribothymidine at the stages of electron relaxation via absorption spectroscopy. Ideally, such experiments should be done in water, which is a more biological solvent, but the relaxation time of the electron is too fast for real-time observation of the solvation steps. However, in cells the lifetime of LEEs should be slower than in pure water and closer to that in DEG. Their spectroscopic results reveal that the quasi-free electron with energy either higher or at the level of conduction band, attaches to ribothymidine. The fast electron capture by ribothymidine leads to a new species characterized in the UV-visible region. Their time-resolved studies are complemented by DFT calculations and analyses of molecular products formed by gamma radiolysis. Considering all the information obtained, the authors assign the transient intermediate state as the excited ribothymidine anion radical that undergoes exclusively an N1-glycosidic bond dissociation rather than relaxation to its ground state.

Response: We thank the reviewer for very positive comments. Not only did the reviewer carefully read our manuscript, he/she grasped the advantage of the present results almost exactly as we described in this manuscript. In the meantime, the reviewer also provided several important suggestions and critiques. We have addressed these concerns and revised our manuscript accordingly. Our detailed responses, as well as new supporting results, are presented below.

1. Experimental results retrieved from the picosecond pulse radiolysis setup.

Within the framework of pump-probe time-resolved absorption spectroscopy, it is well known that measuring appropriately the absorbance of transient species within a spectral range located around and below 400 nm is more than very difficult using the white light generated from sapphire as the probe, mainly because of the very poor

quality of probing light that it produces below 420 nm (3). Furthermore, this point is of particular importance here as the absorbance retrieve at 390 nm is critical to support the authors' line of reasoning. Can the authors explain how they addressed this critical issue to make sure that their results can be trusted?

Such implementation of their pump-probe setup is surprising as the authors seem to be aware of this problem, since they used CaF₂ for the white-light generation, in their previous study (1,2), which is known to appropriately cover the near-UV range.

Response: We agree with the reviewer that the generation of the supercontinuum and the quality of the probe light below 400 nm are of great importance for the reliability of acquired spectral-kinetics data. The electron pump-probe experiments and results reported in our originally submitted manuscript were initially performed at the LINAC facility (Tokyo University); subsequently, these results were reproduced and confirmed at ELYSE facility (University Paris-Sud). The optical design and framework in these two facilities are very similar and the broadband supercontinuum used to generate the white light is CaF₂ crystal at both beamlines, instead of sapphire. We have corrected this error in the description of the experimental setup. At LINAC facility (Japan), the second harmonics (391 nm) and the fundamental light (782 nm) were simultaneously injected into CaF₂; thus, broadband supercontinuum ranging from 370 – 1050 nm with more light around 390 nm was generated and used as the probe light (see Fig. R1). The probe light was detected by a multi-channel analyzer (PMA20, Hamamatsu Photonics Co. Ltd.) which consists of 2048 channels of CCD and covers the wavelength of 350 -1100 nm. The broad absorption band of hydrated (i.e., solvated) electron (e_{sol}^-) was well-established, and it was used as a “reference” to check the reliability of our results from the supercontinuum. As shown in Fig.R1, the spectrum of e_{sol}^- from 370 to 1050 nm measured in water is identical with the reported one (D. M. Bartels et al., *J. Phys. Chem. A* 2005, 109, 1299-1307.)

Figure R1. The supercontinuum generated by CaF₂ at LINAC facility (Left). The transient absorption spectrum of hydrated electron (e_{sol}^-) ranging from 370 to 1050 nm (scattering black points) in comparison with the one in literature (right). e-beam (D = 4mm) and probe light are perpendicularly crossed at the sample cell (1 cm × 1 cm)

to avoid Cherenkov light and to reduce absorption of quartz window as much as possible.

Figure R2. The supercontinuum (generated by CaF₂, ELYSE facility) shape before and after electron pulse (each spectrum is that of the supercontinuum at different time delays of the scan for the reference and the probe channel). The amount of light (at least 2000 counts) is enough for the measurement up to 370 nm.

At ELYSE facility, the supercontinuum, generated by focusing $\sim 1 \mu\text{J}$ of the laser source into a 6-mm-thick CaF₂ disk, was used as the optical probe covering a broad spectral range (370 to 720 nm). Note that the main difference between the two laser-driven electron accelerators is only the energy of the electron pump ($\sim 18 \text{ MeV}$ at LINAC and $\sim 7 \text{ MeV}$ at ELYSE). Concerning the pulse radiolysis experiments using the probe light below 400 nm, we performed two sets of experiments one last June and another one on September 18th, 2018. In experiments carried out on 18th September, the dose was almost two times higher than the one in June and the light is found to be more intense in UV region. In Fig. R2, the shape of the supercontinuum has more than 2000 counts at 370 nm in both cases, which is sufficiently strong and stable to have an accurate detection. As a result, the conclusion is that there is no radiation dose effect. Analyses of both sets at these two different facilities yield very similar results. Consequently, we have revised our experimental part and inserted new Figures (Fig.SI1) representing supercontinuum spectrum in SI for the sake of clarity.

2. Transient absorption spectra analysis.

According to the author's own writing on their previous work (2), the spectra of the type of system studied in the present work are known to strongly overlap; as a result, it becomes difficult to deconvolute individual spectrum. To alleviate this problem, they previously used a global data analysis approach, which seemed very rigorous (section I of the SI of (2)). In the present study, the authors did not specify if the same approach was applied. Is it correct to

conclude that the transient spectra of interest in the present work were retrieved by using a more trivial data analysis method, which appears much less rigorous? If it is actually the case, can the authors explain why they failed in applying the same rigorous data analysis protocol as in (1, 2)?

Response: Some years ago, our colleague - Dr. Pascal Pernot, developed a multivariate curve resolution alternating least squares (MCR-ALS) approach that aims at treating the 2D data shown in Fig. R3 (*J. Photochem. Photobiol. C* **2012**, *13*, 1–27). This program allows us to assess the number of absorbing species in a data matrix and rigorously deconvolute their corresponding kinetics and spectra. It is often useful for the system containing three or more than three absorbing species. In our present work, a combination of both analysis approaches (global MCR-ALS analysis and simple subtraction) have been used to sort out the spectral characterizations of individual species. The analyzed results are consistent, and they lead to the same conclusions.

Figure.R3. 2D Experimental data of pulse radiolysis solution of 300 mM ribothymidine / DEG performed on 18 September (ELYSE facility). The delay line in pump-probe is adjusted from 5 ps to 11 ns.

In the course of the present study, three sets of raw data matrices were acquired at different timescales: one is called “short scan”, the second one is “extended scan”, and the third is “microsecond scan”. In “short scan”, the time delays are limited to 300 ps. The purpose of carrying out pump-probe experiment at these time delays is to investigate the fast electron solvation/attachment. In the “extended scan”, measurements are performed up to 11 ns. The delay line was adjusted with 303 steps to have high resolution (1ps and 2 ps between each points) at a shorter time and 10 ps (middle time) and then 100 ps (at a longer time). Herein by conducting experiments with a solution of 0.3 M ribothymidine (rT), we can obtain the data at the short time showing the decay of excited state of the rT anion radical (TNI*) and at the longer times seeing the formation of ground state of the anion radical (TNI or rT⁻). The complementary measurement called “microsecond scan” was carried out on another accelerator with a flash lamp

as a source of probe light. The goal of this experiment was to study the evolution of TNI (ground state) formed by the solvated electron in a very dilute solution.

Figure. R4. Deconvolution of transient data analysis: (a) spectra of $rT^{\bullet-*}$ and $rT^{\bullet-}$. Inset: kinetics of $rT^{\bullet-*}$ at different concentrations; (b) deconvolution of transient data using MCR-ALS method representing kinetics of $rT^{\bullet-*}$, $rT^{\bullet-}$ and e_{sol}^- .

First, the MCR-ALS analysis of “short scan” and “extended scan” data showed several important features of species (Fig. R4).

- Only 2 species are spectroscopically observed (e_{sol}^- and TNI*) at a short time delay (< 300 ps). The TNI* is formed within the pulse and rapidly decays in a few hundred ps. The decay of the e_{sol}^- is almost negligible. Thus, the ground state of TNI formation through the reaction with e_{sol}^- is not observed at this timescale.
- The third species (TNI, anion radicals) is formed slowly at longer time that correlates with the decay of the e_{sol}^- . The concentration of this species just after the electron pulse is zero (it is not formed within the pulse). The extinction coefficient of this third species should be much lower than that of the e_{sol}^- because its contribution to the data is very weak and the error is more important.
- The solvation of electron leads to the strong changes in the band shape in the red side of the spectrum and disturb the analyzed results. To avoid this and for data analysis of band of electron at the time smaller than 300 ps, we have to limit the wavelength from 370 to 570 nm.

From these above-mentioned discussions, it is apparent that only two species are present in the dataset generated from “short scan”. Thus, the spectral features of these two can be easily distinguished. Besides, as we have shown in Fig.2 in the manuscript, it is emphasized that the presence of ribothymidine (rT) up to 500 mM only decreases the initial infrared absorbance of e_{sol}^- by the reaction of rT with the quasi-free electron (e_{qf}^- (qf = quasi free)) but has little effect on the kinetics of e_{sol}^- at the ps timescale. Consequently, this important point allows us to reasonably and correctly characterize TNI* by subtracting the e_{sol}^- signal as described in following equations (1-3). Accordingly, we can resolve the spectrum of TNI* within our detection spectral window, and more importantly to highlight one of our findings that while e_{qf}^- (qf = quasi free) reacts with rT, the e_{pre}^- does not.

For pulse radiolysis of rT solutions by “short scan”, we could express the absorbance as follows:

$$A(t, \lambda) = A[\text{TNI}^*](t, \lambda) + A[e^-](t, \lambda) \quad 370 \text{ nm} \leq \lambda \leq 1100 \text{ nm}, \quad 0 \leq t \leq 300 \text{ ps} \quad (1)$$

In neat DEG, the absorbance is only associated with e^- :

$$A(t, \lambda) = A[e^-]'(t, \lambda) \quad 370 \text{ nm} \leq \lambda \leq 1100 \text{ nm}, \quad 0 \leq t \leq 300 \text{ ps} \quad (2)$$

As the presence of rT does not change the kinetics of e^- , we could write:

$$[e^-](t, \lambda) = \beta [e^-]'(t, \lambda) \quad 0 < \beta < 1 \quad (3)$$

The parameter β is a constant, representing the ratio of initial infrared absorbance at 15 ps between each rT and neat DEG. The value of β can be found in the curve of Fig. 2d in the text. The loss of e^- signal at 15 ps arises from the e_{qf}^- scavenging.

The decay of TNI* in “short scan” via simple subtraction agree well with the one found in “extended scan” by global analysis and the entire spectrum of TNI* can be reported up to 1100 nm.

To confirm our results obtained in “extended scan” as well as to accurately compare the spectral difference (the band shape, lifetime and extinction coefficient) between TNI* and TNI, we further carried out the “microsecond scan” using rT solutions of very low concentrations (1-10 mM), in which all e^- are converted into e_{sol}^- and TNI* is not formed. In these measurements, when the signal of e_{sol}^- is completely quenched after hundreds of nanoseconds, the spectra of resulting anion radicals (TNI) is clearly observed as the only one absorbing species in the visible region and its lifetime is determinate to be as long as 10 μs .

In summary, we have carefully examined our observations at a wide range of timescales and have applied different strategies of analyses to characterize the species. The results obtained from various analyses are found to be in good agreement.

3. Transient absorption kinetics of the TNI.

According to author’s line of reasoning and the results displayed in Figure 3, the absorption kinetics at 525 nm appear to be of particular interest. Unfortunately, such kinetics are not displayed and properly investigated,

considering that the kinetic of absorbance at 525 nm as a function of time would have provided better insight on the fate of TNIs*, i.e., exclusive decomposition via dissociation of the C-N glycosidic bond versus decay toward TNIs ground state, by analyzing the kinetics at 390 nm versus that at 525 nm. Furthermore, it might have been possible then to evaluate the fraction of excited TNIs that decays toward the ground state of TNIs versus the fraction of TNI* that results in the dissociative process.

Note that, as regards to the dependency of the kinetics displayed in figure 3.b on the concentration of nucleoside, the authors should furnish the characteristic time extract from the exponential fit, together with standard deviations, and let the reader judge if it is independent of ribothymidine concentration.

In figure 1.d, why the kinetics at 1000 nm is not displayed? In figure SI 3, it seems that the absorbance at 1000 nm of 0.3 M ribothymidine / DEG increases relatively to neat DEG at early time (10 ps). Can the authors provide some explanation / interpretation about such results?

Throughout the manuscript and supporting information, display of absorption spectra and kinetics with various different absorbance Y scale / unit, is very confusing and should be made coherent. For example, the values of absorbance associated with the kinetic in 0.3 M riboT displayed in figure 3.b appear inconsistent with the values of absorbance associated with the spectra displayed in figure SI 3.d. While the value of absorbance at 200 ps in figure 3.c matches the one displayed in figure SI 3.d, the absorbance at 40 ps in figure SI 3.d doubles relatively to figure 3.b. The authors need to deal with of such inconsistencies.

Response: We agree with the reviewer's suggestion that the absorption kinetics at around 520 nm is of particular interest. Molecular anion radicals (TNI or $rT^{\bullet-}$) in their ground state display a strong absorbing band at this region, while it is absent in those from excited anion radicals, TNI*. At the beginning of this work, as the reviewer suggested, one of the aims of our work is to estimate the fraction of TNI* dissociation versus that of TNI* relaxation. For clarification, we first estimated the value of molar extinction coefficient (ϵ) of TNI* and TNI, respectively. It was shown that the value of ϵ (TNI) is higher than that of TNI* around 520 nm (see revised Fig.3 in the text). Thus, if the relaxation TNI* occurs at the short time to yield ground state of TNI, we expect that the observed kinetics of $rT^{\bullet-}$ at 520 nm should be much distinct (a slight increase) from those we observed at 390 nm and 600 nm. An increase of TNI from the decay of TNI* is not found (see Fig.R5). We have also deduced the decay of TNI* from kinetics at 520 nm via the same method of analysis and it was found that they are identical with those from 390 nm. Additionally, from the discussion on global analysis mentioned above, it was also learnt the TNI or $rT^{\bullet-}$ signal contribution at the early time is nearly negligible and its formation is only correlated with decay of e_{sol}^- at nanosecond or microsecond timescale. Therefore, it is suggested that the dissociation process of TNI* is dominating and the competitive relaxation is less evident. Due to the higher absorbance in the UV region with better the signal-to-noise ratio, we eventually selected kinetics of TNI* observed at 390 nm to be displayed in our manuscript.

Figure. R5 Transient absorption kinetics and spectra observed in pulse radiolysis of 0.3 M ribothymidine (rT) solution with ELYSE facility.

The fast decay of TNI^* in solutions with various concentrations of rT were normalized to the same absorbance amplitude and it showed that the kinetics are not accelerated or affected by increasing the concentration of rT (see Fig.R6). Thus, the kinetics of TNI^* is thought to be a self-decay that is independent on the yield of TNI^* itself. It is noted that our subtraction approach is not capable to resolve the total decay of TNI^* because at a few nanoseconds, the reaction of e_{sol}^- with rT takes place to produce TNI ; the e_{sol}^- signal cannot be subtracted in this case. In contrast, the completed decay can be seen from global analysis in revised Fig.3 and Fig.R4 and the characteristic time is estimated based on the half-life of TNI^* as well as linear fitting of the averaged kinetics in logarithm. The kinetic of 1000 nm has been added in Fig.1.

According to the Fig.1 and Fig.2 in the manuscript, the initial absorbance at 1000 nm of 0.3 M rT / DEG does not increase relatively to neat DEG at early time (15 ps). In the supplementary information, we have adjusted the ϵ^- absorbance in neat DEG by using the parameter of β to show that how we extracted the spectra of TNI* from the raw data at early time (Fig.S3). This treatment indeed would cause some misunderstanding and confuse the readership on Y scale. As a result, we have carefully checked the inconsistencies and revised the manuscript.

Figure.R6 MCR-ALS analysis of “short scan”, showing only two species is present at early times so that the relaxation pathway of TNI* is less likely to occur at this timescale. Normalized kinetics of TNI* observed in rT solutions are presented to show that the decay of TNI* is very similar with each other and is independent of TNI* yield and the concentration of rT. The linear fitting of absorption kinetics in logarithm estimated a time constant of the TNI* dissociation.

4. Identification of unaltered base (thymine) release via electron-mediated glycosidic bond cleavage in DEA to TNI* of ribothymidine. First, the authors should definitively furnish the mass spectrum of irradiated ribothymidine solution at low concentration (20 mM) to let the reader judge of the absence of thymine. Then, the authors should properly acknowledge the recent study published by Fedor and co-workers (4) that highlights the different outcome of the DEA on hydrated DNA compounds that, unlike DNA compounds under vacuum condition, decompose exclusively via dissociation of the C-N glycosidic bond.

Did the authors made sure that the production of thymine is linear as a function of dose between 0 and 300 kGy? This is of a particular importance to ensure that the base release reaction is a first order process.

The results presented here seems to be based on the assumption that the base release is specific to DEA. However, the release of unaltered bases is a general phenomenon of the free-radical-induced reactions of DNA and its constituents (section 10.8.7 of (5)). By the authors own admission, in the radiolysis of a solution, there occurs numerous reactions between radicals at different timescales from femtoseconds to seconds. The fact that they do not observe the production of thymine at low concentration (20 mM) seems to support their initial hypothesis. But, can the authors explain why the base release induced by the possible reaction of free radicals, e.g. on the sugar moiety, is dismissed?

In my opinion, the author's line of reasoning would be strengthened by providing the quantitative evaluation of thymine production as a function of the concentration of nucleoside to confirm that the kinetic of such production is compatible with the hypothesis of a specific involvement of the DEA process in the base release reaction, and thus establish a stronger correlation between the formation of TNI* and the thymine production.

Response: We have thoroughly performed an additional set of experiments (see in Table SI1) of product analyses of gamma-ray irradiated rT solutions employing ESI-Mass spectroscopy to support the conclusions obtained from pulse radiolysis studies and theoretical calculations. Accordingly, and made revisions in the manuscript and Supplementary information (SI).

As presented in Table R1, a few neat DEG and rT solutions of various concentrations (20, 100, 300 mM) were irradiated by ⁶⁰Co gamma source at different dose up to 335 kGy. The molecular products in each sample were then diluted 100 times by water and measured by ESI-Mass setup at negative mode. The representative mass spectrum of irradiated ribothymidine solution are shown and compared in Figure R7. As can be seen from the figures below and the figures in our first version of SI, the formation of thymine product ($M_w=126$, peak at 125) is observed at the higher concentration (100 and 300 mM), but it is not detected in the solutions of low concentration (20 mM) and solvent (DEG) at radiation does as high as 335 kGy. To evaluate quantitatively the production of thymine as a function of radiation does, we used a set of standard thymine solutions to correlate the mass peak intensity with the concentration. The Fig R8 shows the yield of thymine in 100 mM and 300 mM solution slightly decreases as the dose increases.

These results are not a surprise for us. The observation of thymine peak only at higher concentration (> 50 mM) is in excellent agreement with our picosecond time-resolved pulse radiolysis results that quasi-free electrons (e_{qf}^-) could only be efficiently scavenged or captured at short range. Indeed, as the reviewer pointed out, thymine formation in water solutions can occur via a number of pathways that are reported in the literature, viz. (a) dissociative electron attachment (DEA) of TNI*, (b) probably via the hydroxyl radical ($\cdot OH$) abstracting a H-atom

at C(5) of the sugar moiety followed by ring opening of the sugar, or (c) direct ionization in which the both the spin and charge transfer occur from the excited base cation radical to the sugar moiety at first; subsequently, the deprotonation of the sugar cation radical followed by ring opening of the sugar moiety. The $\cdot\text{OH}$ generated in water radiolysis, is not formed in the radiolysis of alcohols, such as DEG; moreover, the weaker oxidizing alcohol radicals are less likely to attack the sugar sites. Also, the direct effects of radiation are not significant at this concentration range. The absence of the thymine peak at low concentration, in which e_{aq}^- does not react but these other radicals react with target molecule, supported this hypothesis. Therefore, it leads us to conclude the DEA process in DEG is the dominating process for thymine base release in gamma-irradiated solutions of rT.

In irradiated rT solution, many free-radical-induced reactions stemming from ionization of DEG occur at a wide range of timescales. This view is justified because the mass spectrum of irradiated DEG (see in Fig R7a) shows that quite a few organic molecules are formed via radiolysis of the neat solvent (i.e., DEG) only. Dissociative electron attachment process is one of the earliest process and it leads to thymine formation at picosecond timescale via a first-order decay of excited states of anion radicals, TNI^* . We cannot eliminate the possibilities that thymine will be subsequently transformed to other molecular derivatives through secondary radicals' reactions taking place at microsecond or millisecond. Therefore, the correlation between the ultimate productions of thymine with the radiation dose or the formation of TNI^* observed at picosecond would not be an essential point to support our time-resolved results. On the other hand, the fact that the thymine yield slightly decreases at an increasing dose suggests free-radicals may continue to induce the consumption of thymine products via secondary reactions. Regardless the complexity in radiolysis of the liquids, we successfully identified the formation of thymine which is an evidence of the occurrence of base release via DEA process in liquid DEG containing a significant number of solute molecules.

Fedor and co-workers (*J. Phys. Chem. Lett.*, 2016, 7 (17), pp 3401–3405) recently used a “bottom-up” approach in a molecular beam, by progressively micro-hydrating the target molecule. This study highlights the suppression of dissociation of the C-N glycosidic bond by increasing the hydration degree of thymine and uracil. Their conclusion, from another point of view, is consistent with present study that DEA occur efficiently at less hydrated molecule condition. We appreciate their work because it provided some new insights in understanding the role of water played in dissociative electron attachment. We discussed their findings in our previous paper and we have cited and acknowledged this important work again in our revised manuscript.

Table R1. The list of ribothymidine nucleoside solutions that were subjected to gamma radiation. All samples (solutions) in the tubes were bubbled at least 15 min with Argon and then they were sealed prior to radiation. Note: Gy is a unit of radiation dose, 1 Gy = 1 J/kg. Irradiated volume: 10 mL for each sample.

Samples 1	1-0	1-1	1-2	1-3	1-4	1-5	1-6
Neat DEG	0 kGy	30 kGy	110 kGy	140 kGy	195 kGy	225 kGy	335 kGy
Samples 2	2-0	2-1	2-2	2-3	2-4	2-5	2-6
20 mM rT	0 kGy	30 kGy	110 kGy	140 kGy	195 kGy	225 kGy	335 kGy
Samples 3	3-0	3-1	3-2	3-3	3-4	3-5	3-6
100 mM rT	0 kGy	30 kGy	110 kGy	140 kGy	195 kGy	225 kGy	335 kGy
Samples 4	4-0	4-1	4-2	4-3	4-4	4-5	4-6
300 mM rT	0 kGy	30 kGy	110 kGy	140 kGy	195 kGy	225 kGy	335 kGy

Figure. R7 (a) Mass spectrum of irradiated neat diethylene glycol (DEG). (b) Mass spectrum of irradiated 20 mM ribothymidine in DEG. (c) Mass spectrum of irradiated 100 mM ribothymidine in DEG. (d) Mass spectrum of irradiated 300 mM ribothymidine in DEG. The spectra are obtained by using LC/MS mass spectrometer (Thermo Fisher Scientific) at negative ion mode. The mobile phase uses 100% ethanol at a flow rate of 0.2 mL/min.

Figure. R8 Left: A set of standard thymine solutions used in ESI-Mass measurements to establish the correlation between $m/z = 125$ peak intensity with thymine concentration. Right: the plot of the peak intensity as a function of radiation dose. 300 mM rT in red and 100 mM rT and in blue, respectively.

Minor comments:

I suggest that the word “direct” be deleted from the title. After all the data manipulation to measure the dynamics of the states of the electrons, most readers may not see these measurements as very direct. On the other hand, the authors may want to mention in the title that the measurements were performed in the liquid state.

Response: Indeed, we have mentioned in the title that the measurements were performed in the liquid state, but the title suitable for Nature communication should be 15 words or fewer. Thus, we used “quasi-free electron”, implying that the measurements are done in liquid phase. Also, it specified the states of weakly bound electron for the first time that could break the chemical bond and it might also be of interest to a broader readership such as those who are working on electron/charge transfer or semiconductors.

Following the suggestion of the reviewer, we have modified the title of our paper as:

“Quasi-free electron attachment to nucleoside in solution: excited anion radical formation and its dissociation”

I generally found the manuscript to be difficult to read. The authors should carefully review the next version of the manuscript. The authors need also to be careful about typographical errors, e.g. errors in the legend of figure SI 3.c, in the Y-label of figure SI 3.d, in the conclusion of the manuscript.

[Answer]

We have carefully corrected those errors and we invited the native English-speaking colleagues to copy-edit our manuscript.

Line 51 : write “in gas- and condense-phase”

Response: We have addressed this point.

Lines 52-55 The sentence “The low-energy resonance features in the yield of DSBs, SSBs, and anions produced by the impact of LEEs on model pyrimidine bases suggested that the initial step involves electron capture into the LUMOs (lowest unoccupied molecular orbitals) of nucleobases, creating excited states of transient negative ions (TNIs*)” is ambiguous. Electron capture in lowest unoccupied molecular orbital cannot produce a TNI in an excited state, but they write “orbitals”. How many orbitals are considered lowest?

Response: We have revised it as follows: “The low-energy resonance features in the yield of DSBs, SSBs, and anions produced by the impact of LEEs on model pyrimidine bases suggested that the initial step involves electron capture into the unoccupied molecular orbitals that are above the LUMOs (lowest unoccupied molecular orbitals) of the parent nucleobase, creating excited transient negative ions (TNIs*)”.

Lines 61-64: Here the authors introduce the different states of the electron with the sentences "In a polar medium (e.g., water), the most relevant phase of biology and chemistry, LEEs successively lose energy to become quasi-free electrons (e_{qf}^-) and they can undergo multistep solvation prior to their complete localization as $esol^-$. The transition (e_{pre}^-) from e_{qf}^- to $esol^-$ is accompanied by the appearance of.....". They refer to (e_{pre}^-), e_{qf}^- and $esol^-$, but these symbols are not defined yet. I suggest that they first define the different states of the electron with their energy diagram of Scheme. 1, including that of LEEs.

Response: In the revised version, we have made a definition of states of electron including e_{pre}^- , e_{qf}^- and $esol^-$ and LEEs in Scheme.1.

Recommendation: Publication in Nature Communications, after the authors have adequately answered all comments in this report.

References

- (1) J. Ma, F. Wang, S. A. Denisov, A. Adhikary, and M. Mostafavi, *Science Advances* 3 (2017), 10.1126/sciadv.1701669, <http://advances.sciencemag.org/content/3/12/e1701669.full.pdf>.
- (2) J. Ma, S. A. Denisov, J.-L. Marignier, P. Pernot, A. Adhikary, S. Seki, and M. Mostafavi, *The Journal of Physical Chemistry Letters* 9, 5105 (2018), PMID: 30132673, <https://doi.org/10.1021/acs.jpcclett.8b02170>.
- (3) M. Bradler, P. Baum, and E. Riedle, *Applied Physics B* 97, 561 (2009).
- (4) J. Kocisek, B. Sedmidubská, S. Indrajith, M. Farnik, and J. Fedor, *The Journal of Physical Chemistry B* 122, 5212 (2018), PMID: 29706064, <https://doi.org/10.1021/acs.jpcc.8b03033>.
- (5) C. von Sonntag, *Free-Radical-Induced DNA Damage and Its Repair* (Springer-Verlag Berlin Heidelberg, 2006).

Reviewer #2 (Remarks to the Author):

The authors use pico-second laser-driven electron pulse radiolysis to study dissociative electron attachment (DEA) to selected DNA components in real liquid environments under ambient conditions. They determine that the intermediate is an electronically-excited anion which rapidly decays by DEA into products. On the one hand, I do not find this especially surprising, but on the other hand, this appears to be the first actual demonstration of that process in a real liquid. While parts of this manuscript are well written, other parts would be improved if a native English speaker were to edit it. On balance I like this manuscript and feel that it presents important new findings. I support its publication in Nature Communications.

Response:

We appreciate for the positive comments. The English of manuscript was corrected and significantly improved by our native English-speaking colleagues.

Reviewers' comments:

Reviewer #1 (Remarks to the Author):

Review of manuscript NCOMMS-18-26474A:

Quasi-free electron attachment to nucleoside in solution: excited anion radical formation and its dissociation.

First, let me acknowledge that the present resubmitted manuscript is notably improved. The authors have considered all of my comments and corrections. They did additional experiments to provide answers and check the validity of their assertions. Most of my concerns as regards the previous version of the manuscript have been addressed rigorously. I have rarely seen such an impressive improvement in a manuscript.

At this point, only the study of the identification of unaltered base (thymine) release via electron-mediated glycosidic bond cleavage in TNI* of ribothymidine remains problematic. Mainly, the issues with the results presented here come with the huge dose of ionizing radiation used during this investigation. As mentioned by the authors, irradiating with such an amount of dose involves the occurrence of less likely events, such as the secondary reactions between the radio-induced byproducts of DEG and rT. Within this framework, it is difficult to acknowledge the authors' argument in favor of the negligible contribution of a first order process resulting in the base (thymine) release, due to the reaction between the reactive species (free radicals) produced upon the radiolysis of DEG and the native rT. I am wondering if the author had no choice to use such an amount of dose to be able to detect the production of Thymine, because of the limitation (sensitivity) of the technique used in their study (ESI-LC/MS). In my opinion, it would have been preferable to restrain the analyses to a regime of irradiation at lower total dose exposure, where the production kinetic of thymine as a function of dose is linear and the secondary reactions between the radio-induced byproducts of DEG and the thymine remain negligible, by using more sensitive quantitative analysis techniques (e.g. LC-MS/MS). Then, the authors would have been able to investigate if the production kinetics of thymine matches their hypothesis. Indeed, even if the absence of thymine at low concentration (20 mM) versus its presence at higher concentration (100 mM and 300 mM) agrees with the authors hypothesis, observation of a very similar quantity of thymine produced in the 100 mM and 300 mM solutions seems to refute the same hypothesis and thus appears here very problematic.

I suggests that, either the corresponding part of the study be redone properly and the results included in the present manuscript, as it need to be completed with stronger evidence, or the corresponding section be removed from the final manuscript before publication, as I consider that a revised manuscript, without the problematic part, is still worthy of publication in Nature communication. In conclusion, the authors provide sufficient evidence to demonstrate that DEA to a biological molecule occurs in the liquid phase, likely giving rise to base release. Consequently, the paper should be published in Nature communication.

Minor comment:

Using a super-continuum probe light that extend only to ~370 nm in the blue range (figure SI.1), the authors should limit the display of the spectra to 370 nm in their manuscript and certainly not extend it to 350 nm (e.g. figure 3.a).

Response to reviewers' comments

Reviewer: 1

First, let me acknowledge that the present resubmitted manuscript is notably improved. The authors have considered all of my comments and corrections. They did additional experiments to provide answers and check the validity of their assertions. Most of my concerns as regards the previous version of the manuscript have been addressed rigorously. I have rarely seen such an impressive improvement in a manuscript.

Response: We are pleased by the encouragement provided by the reviewer.

At this point, only the study of the identification of unaltered base (thymine) release via electron-mediated glycosidic bond cleavage in TNI* of ribothymidine remains problematic. Mainly, the issues with the results presented here come with the huge dose of ionizing radiation used during this investigation. As mentioned by the authors, irradiating with such an amount of dose involves the occurrence of less likely events, such as the secondary reactions between the radio-induced byproducts of DEG and rT. Within this framework, it is difficult to acknowledge the authors' argument in favor of the negligible contribution of a first order process resulting in the base (thymine) release, due to the reaction between the reactive species (free radicals) produced upon the radiolysis of DEG and the native rT. I am wondering if the author had no choice to use such an amount of dose to be able to detect the production of Thymine, because of the limitation (sensitivity) of the technique used in their study (ESI-LC/MS). In my opinion, it would have been preferable to restrain the analyses to a regime of irradiation at lower total dose exposure, where the production kinetic of thymine as a function of dose is linear and the secondary reactions between the radio-induced byproducts of DEG and the thymine remain negligible, by using more sensitive quantitative analysis techniques (e.g. LC-MS/MS). Then, the authors would have been able to investigate if the production kinetics of thymine matches their hypothesis. Indeed, even if the absence of thymine at low concentration (20 mM) versus its presence at higher concentration (100 mM and 300 mM) agrees with the authors hypothesis, observation of a very similar quantity of thymine produced in the 100 mM and 300 mM solutions seems to refute the same hypothesis and thus appears here very problematic.

I suggest that, either the corresponding part of the study be redone properly and the results included in the present manuscript, as it needs to be completed with stronger evidence, or the corresponding section be removed from the final manuscript before publication, as I consider that a revised manuscript, without the problematic part, is still worthy of publication in Nature communication. In conclusion, the authors provide sufficient evidence to demonstrate that DEA to a biological molecule occurs in the liquid phase, likely giving rise to base release. Consequently, the paper should be published in Nature communication.

Response: Following the critique of this reviewer, we have removed the data and their discussions on product analysis from the main manuscript and also from the Supporting Information completely.

Minor comment:

Using a super-continuum probe light that extends only to ~370 nm in the blue range (figure SI.1), the authors should limit the display of the spectra to 370 nm in their manuscript and certainly not extend it to 350 nm (e.g. figure 3.a).

Response: This has been addressed.